# *Top Leaderboard Ranking = Top Coding Proficiency, Always?* 🦋 EVOEVAL: Evolving Coding Benchmarks via LLM

**Chunqiu Steven Xia**[*]   **Yinlin Deng**[*]                    **Lingming Zhang**

University of Illinois Urbana-Champaign 🔶

{chunqiu2, yinlind2, lingming}@illinois.edu

## Abstract

Large language models (LLMs) have become the go-to choice for code generation tasks, with an exponential increase in the training, development, and usage of LLMs specifically for code generation. To evaluate the ability of LLMs on code, both academic and industry practitioners rely on popular handcrafted benchmarks. However, prior benchmarks contain only a very limited set of problems, both in quantity and variety. Further, due to popularity and age, many benchmarks are prone to data leakage where example solutions can be readily found on the web and thus potentially in training data. Such limitations inevitably lead us to inquire: *Is the leaderboard performance on existing benchmarks reliable and comprehensive enough to measure the program synthesis ability of LLMs?* To address this, we introduce 🦋 EVOEVAL– a program synthesis benchmark suite created by *evolving* existing benchmarks into different targeted domains for a comprehensive evaluation of LLM coding abilities. Our study on **57** LLMs shows that compared to the high performance obtained on standard benchmarks like HUMANEVAL, there is a significant drop in performance (on average 38.1%) when using EVOEVAL. Additionally, the decrease in performance can range from 19.6% to 47.7%, leading to drastic ranking changes amongst LLMs and showing potential overfitting of existing benchmarks. Furthermore, we showcase various insights including the brittleness of instruction-following models when encountering rewording or subtle changes as well as the importance of learning problem composition and decomposition. EVOEVAL not only provides comprehensive benchmarks, but can be used to further evolve arbitrary problems to keep up with advances and the ever-changing landscape of LLMs for code. We have open-sourced our benchmarks, tools, and all LLM-generated code at https://github.com/evo-eval/evoeval.

## 1 Introduction

Program synthesis (Gulwani et al., 2017) is regarded as the *holy-grail* in the field of computer science. Recently, large language models (LLMs) have become the default choice for program synthesis due to its code reasoning capabilities acquired through training on code repositories. Popular LLMs like GPT-4 (OpenAI, 2023), Claude-3.5 (Anthropic, 2024b), and Gemini (Team et al., 2023) have shown tremendous success in aiding developers on a wide range of coding tasks (Chen et al., 2021; Xia & Zhang, 2023; Deng et al., 2023b). Furthermore, researchers and practitioners have designed code LLMs (e.g., DeepSeek Coder (Guo et al., 2024), CodeLlama (Rozière et al., 2023), and StarCoder (Li et al., 2023)) using a variety of training methods designed specifically for the code domain to improve LLM code understanding.

Coding benchmarks like HUMANEVAL (Chen et al., 2021) and MBPP (Austin et al., 2021) have been handcrafted to evaluate the program synthesis task of turning natural language descriptions (e.g., docstrings) into code snippets. These code benchmarks measure

---

[*]Contributed equally with author ordering decided by *Nigiri*.

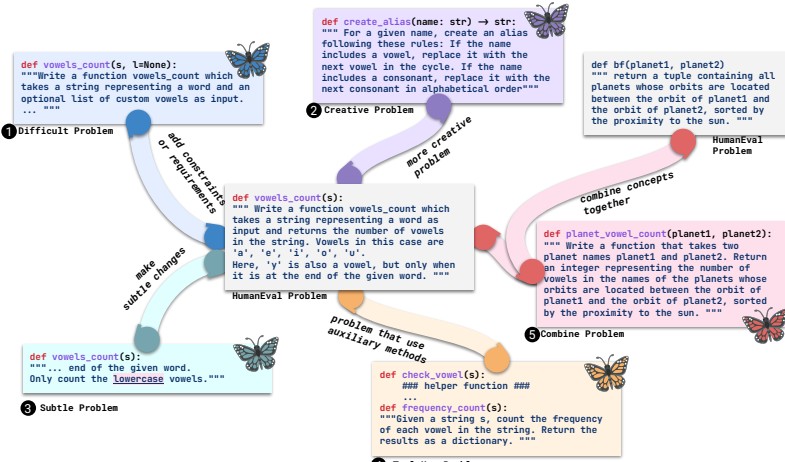

Figure 1: Exemplar problems generated in EVOEVAL starting from a HUMANEVAL problem.

functional correctness by evaluating LLM-generated solutions against a set of limited predefined tests. Recent work (Liu et al., 2023) has further included augmented tests to rigorously evaluate the functional correctness of LLM generated code. However, apart from test inadequacy, existing popular code synthesis benchmarks have the following limitations:

- **Limited amount and variety of problems.** Code benchmarks are mainly constructed by human annotators manually. Due to the high manual effort required, they only contain a limited amount of problems (e.g., only 164 problems in HUMANEVAL). Such a low amount of problems is not sufficient to fully measure the complete spectrum of program synthesis capability of LLMs. Additionally, prior benchmarks include mostly self-contained problems that lack variety in both types and domains, where the final evaluation output only shows the percentage of problems solved. While they provide a baseline overview of the coding abilities, LLM builders and users cannot gain deeper insights to exactly which problem types or coding scenarios the particular LLM may excel in or struggle with.
- **Prone to data leakage and training dataset composition.** Popular benchmarks like HUMANEVAL and MBPP were released almost 4 years ago, with example solutions available in third-party open-source repositories. In fact, recent work (Riddell et al., 2024) has shown that there is substantial overlap between benchmark solutions and open-source training corpuses. Furthermore, the problems within these benchmarks are often simple derivatives of common coding problems. While recent LLMs have been climbing the leaderboard by achieving higher pass@1 scores (often with minimal difference between the next best model), it is unclear whether high scores achieved by LLMs are truly due to their learned coding capability or instead obtained via memorizing benchmark solutions.

As more LLMs are being constructed, trained, and used especially for code, the insufficient evaluation benchmarks raise a critical question: *Is leaderboard performance on existing benchmarks reliable and comprehensive enough to measure the program synthesis ability of LLMs?*

**Our work.** To address the limitation of existing benchmarks, we introduce 🦋 EVOEVAL – a set of program synthesis benchmarks created by *evolving* existing problems. The key idea behind EVOEVAL is to use LLMs to automatically transform existing problems into targeted domains, enabling more comprehensive evaluations. Unlike prior benchmark construction approaches, which either obtain problems from open-source repositories (posing data leakage risks) or require manual construction (resulting in high manual effort and limited diversity), EVOEVAL leverages LLMs with targeted transformations to synthesize new coding problems. Specifically, we design five such transformations: *Difficult, Creative, Subtle, Combine, and Tool Use*. We then prompt GPT-4 to independently transform any existing problem in previous benchmarks into a new problem within the targeted domain.

Figure 1 shows a concrete example of EVOEVAL starting with an initial problem in HU-MANEVAL– `vowels_count` to count the number of vowels. ❶ We first observe the transformation to a more difficult problem by asking GPT-4 to add additional requirements. This new

problem contains a separate custom vowel list that makes the overall program logic more complex. ❷ We can craft a more creative problem of `create_alias` that still uses concepts like vowels and consonants but involves a much more innovative and unusual problem description. ❸ We can also make subtle changes to the problem where we only count the lowercase vowels to test if the LLM is simply memorizing the benchmark. ❹ We can additionally combine concepts from multiple problems together. In the example, we use another problem `bf` to create a new problem that returns the vowels in each planet sorted based on the orbiting order. ❺ Furthermore, we can test LLMs' ability to utilize helper functions (commonplace in real-world code repositories) to solve more complex problems. Again, we reuse the concepts of vowels from the initial problem. However, instead of directly solving the problem, the LLM can use the provided `check_vowel` helper function to simplify the solution.

Together, these transformed benchmarks are designed to introduce more challenging problems and assess different aspects of LLMs' code understanding and synthesis abilities. In EVOEVAL, we additionally use GPT-4 to generate the ground truth solution to each problem as well as rigorous test cases to evaluate the functional correctness of LLM-synthesized code. Finally, we manually check each generated problem and ground truth to ensure problem clarity and correctness. EVOEVAL serves as a way to further evolve existing benchmarks into more complex and well-suited problems for evaluation in order to keep up with the ever-growing LLM research. Our work makes the following contributions:

- **Benchmarks**: We present EVOEVAL– a set of program synthesis benchmarks created by evolving HUMANEVAL problems. EVOEVAL includes 828 problems across 7 benchmarks, equipped with ground truth solutions and test cases to evaluate functional correctness.
- **Approach**: We propose a complete pipeline to generate new coding problems for benchmarking by evolving existing problems through targeted transformations via LLMs. Furthermore, our pipeline reduces manual checking effort by automatically refining problem inconsistencies, generating ground truth, and producing test cases.
- **Study**: We conduct a comprehensive study on **57** LLMs. We found that compared to the high performance on prior benchmarks, LLMs significantly drop in accuracy (average 38.1%) on EVOEVAL. Additionally, this drop is not uniform across LLMs (from 19.6% to 47.7%), leading to drastic ranking changes. We further demonstrate that certain LLMs cannot keep up their high performance when evaluated on more challenging tasks or problems in different domains, highlighting the possibility of overfitting to existing benchmarks. Moreover, we observe that instruction-following LLMs are sensitive to rephrasing or subtle changes in the problem description. They also struggle with utilizing already provided auxiliary functions. We further demonstrate that current LLMs fail to effectively compose multiple general coding concepts to solve more complex variants, or address subproblems decomposed from problems they previously solved.

## 2 Approach

**Targeted problem transformation.** We first prompt a powerful LLM to evolve an existing problem into a new one using a transformation prompt. Each transformation prompt aims to transform the existing problem in a specific manner. We define two different transformation types: **semantic-altering** – changes the semantic meaning of the problem and **semantic-preserving** – modifies the description while keeping the same semantic meaning.

**Problem refinement & ground truth generation.** The initial evolved problem produced by the LLM may include inconsistencies like incorrect examples. For coding benchmarks, such mistakes can lead to inaccurate evaluation. As such, we introduce a refinement pipeline to iteratively rephrase and refine the problem as needed. We first query the LLM to obtain a possible solution and test inputs for the initial problem. We then evaluate the test inputs on the solution to derive the expected outputs. Next, we instruct the LLM to refine the problem by adding or fixing the example test cases in the docstring using the computed test inputs/outputs, and then regenerate a solution. We then check if the new solution on the test inputs produces the same outputs as the previous solution. The intuition is that since the refined problem should only include minimal changes, the solution output should then remain the same in the absence of any inconsistencies. As such, if we observe

differences between the two solution outputs, we ask the LLM to further revise and fix any inconsistencies and repeat the process. If both solutions agree on outputs, we return the new problem description, solution, and test cases for functional evaluation.

**Manual examination & test augmentation.** For each transformed problem, we carefully examine and adjust any final faults to ensure each problem and ground truth are correctly specified and implemented. We further generate additional tests using an LLM-based test augmentation technique (Liu et al., 2023). Finally, we produce EVOEVAL, a comprehensive code synthesis benchmark suite containing diverse problems to evaluate LLM coding capability across various domains. Details like transformation prompts are presented in Appendix D.

## 3 EVOEVAL Benchmarks & Evaluation Methodology

EVOEVAL uses HUMANEVAL problems as seeds and GPT-4 as the foundation LM to produce 828 problems across 7 different benchmarks (5 semantic-altering and 2 semantic-preserving). For the semantic-altering benchmarks, we generate 100 problems each using different seed problems from HUMANEVAL. For the semantic-preserving benchmarks, we transform all 164 problems in HUMANEVAL as we reuse the original ground truths, requiring less validation.

- **DIFFICULT**: Increase complexity by adding constraints, replacing commonly used requirements to less common ones, or introducing additional steps to the original problem.
- **CREATIVE**: Produce a more creative problem using stories or narratives.
- **SUBTLE**: Make a subtle change such as inverting or replacing a requirement.
- **COMBINE**: Combine two problems by using concepts from both problems.
- **TOOL_USE**: Produce a main problem and helper functions. Each helper function is fully implemented and provides hints or useful functionality for solving the main problem.
- **VERBOSE**: Reword the original docstring to be more verbose with descriptive language
- **CONCISE**: Reword the original docstring to be more concise using concise language.

**Evaluation setup:** Each LLM generated sample is executed against the test cases and evaluated using differential testing (McKeeman, 1998) – comparing against the ground truth results to measure functional correctness. We focus on greedy decoding and denote this as pass@1.

**Models:** We evaluate **57** LLMs (Appendix C), including both proprietary and open-source models. Further, we classify the LLMs as either base or instruction-following and discuss the effect of model variants.

**Input format:** To produce the code solution using each LLM, we provide a specific input prompt: For base LLMs, we let the LLM autocomplete the solution given the function header and docstring. For instruction-following LLMs, we use the recommended instruction and ask the LLM to generate a complete solution for the problem.

## 4 Results

### 4.1 LLM Synthesis & Evaluation on EVOEVAL

**EVOEVAL produces more complex and challenging benchmarks for program synthesis.** Table 1 shows the pass@1 performance along with the ranking of LLMs on each of the semantic-altering EVOEVAL benchmarks with the average pass@1 and ranking on all benchmarks in the last columns[1]. First, compared to the success rate on HUMANEVAL, when evaluated on EVOEVAL, all LLMs **consistently perform worse**. For example, the state-of-the-art GPT-4o, GPT-4 and Claude-3.5 models solve close to 85% of all HUMANEVAL problems but fall almost below 55% pass@1 when evaluated on the DIFFICULT problems. On average, across all benchmarks, the performance of LLMs decreased by 38.1% (DIFFICULT: 56.6%, CREATIVE: 48.2%, SUBTLE: 5.0%, COMBINE: 74.7%, and TOOL_USE: 6.1%). Additionally, this drop is not uniform across all LLMs and can range from 19.6% to 47.7%.

---

[1]We evaluated all 57 LLMs, however, we omitted some LLMs in Table 1 for space reasons.

Table 1: pass@1 and ranking results (* indicates tie) on the semantically-altering EVOEVAL and HUMANEVAL benchmarks (including HUMANEVAL+ scores in the parenthesis). 💬 denotes instruction-following LLMs.

| | Size | HUMANEVAL | | 🦋DIFFICULT | | 🦋CREATIVE | | 🦋SUBTLE | | 🦋COMBINE | | 🦋TOOL_USE | | 🦋EVOEVAL | |
|---|---|---|---|---|---|---|---|---|---|---|---|---|---|---|---|
| | | pass@1 | rank | pass@1 | rank | pass@1 | rank | pass@1 | rank | pass@1 | rank | pass@1 | rank | pass@1 | rank |
| GPT-4o | NA | 86.0 (81.7) | **1** | 51.0 | 5 | 64.0 | **2** | 80.0 | 4 | 51.0 | *3 | 72.0 | 1 | 67.3 | **2** |
| GPT-4-Turbo | NA | 83.5 (80.5) | *2 | 50.0 | *6 | 61.0 | **3** | 82.0 | *1 | 45.0 | 5 | 69.0 | *3 | 65.1 | 4 |
| GPT-4 | NA | 82.3 (76.2) | *6 | 52.0 | *2 | 66.0 | **1** | 76.0 | 6 | 53.0 | **2** | 68.0 | *6 | 66.2 | **3** |
| GPT-3.5-Turbo | NA | 76.8 (69.5) | *9 | 33.0 | *19 | 42.0 | *13 | 70.0 | *9 | 33.0 | 9 | 64.0 | *10 | 53.1 | 12 |
| Claude-3.5 | NA | 83.5 (78.0) | *2 | 56.0 | 1 | 60.0 | 4 | 82.0 | *1 | 57.0 | 1 | 68.0 | *6 | 67.8 | 1 |
| Claude-3 | NA | 82.3 (75.0) | *6 | 50.0 | *6 | 53.0 | 7 | 81.0 | **3** | 42.0 | 7 | 69.0 | *3 | 62.9 | 7 |
| Claude-3-haiku | NA | 74.4 (66.5) | *13 | 40.0 | *12 | 47.0 | *11 | 65.0 | *16 | 25.0 | *12 | 61.0 | *16 | 52.1 | 13 |
| Claude-2 | NA | 66.5 (62.2) | *24 | 29.0 | 23 | 42.0 | *13 | 64.0 | *19 | 19.0 | 20 | 57.0 | *22 | 46.2 | 21 |
| Gemini-1.5-pro | NA | 83.5 (76.8) | *2 | 52.0 | *2 | 55.0 | *5 | 78.0 | 5 | 43.0 | 6 | 69.0 | *3 | 63.4 | 6 |
| Gemini-1.0-pro | NA | 62.2 (56.7) | 27 | 37.0 | *16 | 40.0 | 18 | 53.0 | *27 | 23.0 | *15 | 57.0 | *22 | 45.4 | 23 |
| PaLM-2 | NA | 40.2 (36.6) | 44 | 18.0 | *38 | 22.0 | 39 | 36.0 | *48 | 3.0 | *45 | 46.0 | *35 | 27.5 | 43 |
| DS Coder-v2-Inst | 236b | 82.9 (78.7) | 5 | 52.0 | *2 | 55.0 | *5 | 75.0 | 7 | 51.0 | *3 | 70.0 | 2 | 64.3 | 5 |
| DS Coder-Inst | 33b | 78.0 (73.2) | 8 | 47.0 | 9 | 47.0 | *11 | 67.0 | *11 | 31.0 | *10 | 66.0 | 8 | 56.0 | 8 |
| | 6.7b | 74.4 (69.5) | *13 | 40.0 | *12 | 37.0 | *19 | 61.0 | *23 | 18.0 | *21 | 51.0 | 30 | 46.9 | 20 |
| | 1.3b | 63.4 (60.4) | 26 | 20.0 | *36 | 25.0 | *31 | 53.0 | *27 | 9.0 | *34 | 39.0 | *47 | 34.9 | 30 |
| DS Coder | 33b | 50.6 (42.7) | 32 | 26.0 | 26 | 23.0 | *36 | 47.0 | *32 | 11.0 | *31 | 63.0 | *13 | 36.8 | 29 |
| | 6.7b | 45.1 (38.4) | *37 | 21.0 | *32 | 24.0 | *33 | 47.0 | *32 | 5.0 | *41 | 55.0 | *25 | 32.9 | 35 |
| | 1.3b | 29.9 (26.2) | 51 | 6.0 | *54 | 19.0 | *41 | 27.0 | 55 | 0.0 | 57 | 40.0 | 46 | 20.3 | 51 |
| DS Coder-1.5-Inst | 7b | 68.9 (63.4) | *21 | 37.0 | *16 | 37.0 | *19 | 66.0 | *14 | 24.0 | 14 | 60.0 | *18 | 48.8 | 16 |
| DS Coder-1.5 | 7b | 42.1 (34.8) | *41 | 21.0 | *32 | 34.0 | *23 | 43.0 | *37 | 4.0 | *43 | 54.0 | *27 | 33.0 | 34 |
| Llama -3.1-Inst | 70b | 75.0 (68.9) | *11 | 42.0 | 10 | 49.0 | 9 | 73.0 | 8 | 34.0 | 8 | 62.0 | 15 | 55.8 | 9 |
| Llama -3-Inst | 70b | 73.8 (71.3) | *15 | 41.0 | 11 | 52.0 | 8 | 70.0 | *9 | 31.0 | *10 | 64.0 | *10 | 55.3 | 10 |
| CodeLlama-Inst | 70b | 66.5 (59.8) | *24 | 31.0 | 22 | 41.0 | *16 | 65.0 | *16 | 18.0 | *21 | 65.0 | 9 | 47.7 | 18 |
| | 34b | 51.8 (43.9) | 31 | 22.0 | *30 | 27.0 | 29 | 43.0 | *37 | 9.0 | *34 | 47.0 | *33 | 33.3 | 33 |
| | 13b | 48.8 (42.7) | 35 | 21.0 | *32 | 25.0 | *31 | 46.0 | 35 | 8.0 | *37 | 54.0 | *27 | 33.8 | 32 |
| | 7b | 43.3 (39.0) | 39 | 14.0 | 44 | 18.0 | *43 | 40.0 | *43 | 8.0 | *37 | 44.0 | *39 | 27.9 | 42 |
| CodeLlama | 70b | 60.4 (52.4) | 29 | 25.0 | 27 | 29.0 | *26 | 49.0 | *29 | 14.0 | *27 | 63.0 | *13 | 40.1 | 28 |
| | 34b | 52.4 (43.3) | 30 | 15.0 | 43 | 24.0 | *33 | 47.0 | *32 | 11.0 | *31 | 44.0 | *39 | 32.2 | 36 |
| | 13b | 42.7 (36.6) | 40 | 18.0 | *38 | 24.0 | *33 | 38.0 | *45 | 6.0 | 40 | 48.0 | *31 | 29.4 | 39 |
| | 7b | 39.6 (36.6) | 45 | 10.0 | *48 | 15.0 | 47 | 42.0 | 40 | 3.0 | *45 | 44.0 | *39 | 25.6 | 44 |
| WizardCoder | 34b | 61.6 (54.3) | 28 | 24.0 | 28 | 32.0 | 25 | 55.0 | 26 | 17.0 | *24 | 55.0 | *25 | 40.8 | 26 |
| WizardCoder-1.1 | 33b | 73.8 (69.5) | *15 | 48.0 | 8 | 48.0 | 10 | 66.0 | *14 | 20.0 | 19 | 64.0 | *10 | 53.3 | 11 |
| XwinCoder | 34b | 68.9 (62.2) | *21 | 33.0 | *19 | 42.0 | *13 | 67.0 | *11 | 15.0 | 26 | 60.0 | *18 | 47.7 | 19 |
| Phind-CodeLlama-2 | 34b | 70.7 (66.5) | 19 | 22.0 | *30 | 35.0 | 22 | 63.0 | 21 | 25.0 | *12 | 58.0 | 21 | 45.6 | 22 |
| Code Millenials | 34b | 73.2 (69.5) | 17 | 35.0 | 18 | 41.0 | *16 | 65.0 | *16 | 17.0 | *24 | 56.0 | 24 | 47.9 | 17 |
| Speechless-CL | 34b | 75.0 (69.5) | *11 | 38.0 | 15 | 37.0 | *19 | 64.0 | *19 | 23.0 | *15 | 59.0 | 20 | 49.3 | 15 |
| Magicoder-s-DS | 6.7b | 76.8 (70.7) | *9 | 40.0 | *12 | 34.0 | *23 | 67.0 | *11 | 21.0 | *17 | 61.0 | *16 | 50.0 | 14 |
| Magicoder-s-CL | 7b | 70.1 (65.9) | 20 | 27.0 | 25 | 26.0 | 30 | 58.0 | 25 | 11.0 | *31 | 52.0 | 29 | 40.7 | 27 |
| StarCoder2 | 15b | 45.1 (36.0) | *37 | 16.0 | *41 | 19.0 | *41 | 41.0 | *41 | 5.0 | *41 | 48.0 | *31 | 29.0 | 41 |
| | 7b | 34.8 (31.1) | *46 | 12.0 | *45 | 17.0 | 45 | 38.0 | *45 | 2.0 | *51 | 46.0 | *35 | 25.0 | 46 |
| | 3b | 31.1 (26.2) | 50 | 8.0 | *51 | 14.0 | *48 | 31.0 | *50 | 2.0 | *51 | 35.0 | 51 | 20.2 | 52 |
| StarCoder | 15b | 34.8 (30.5) | *46 | 12.0 | *45 | 11.0 | 53 | 37.0 | 47 | 2.0 | *51 | 44.0 | *39 | 23.5 | 47 |
| Mixtral-Inst | 8x7b | 42.1 (38.4) | *41 | 21.0 | *32 | 18.0 | *43 | 41.0 | *41 | 9.0 | *34 | 45.0 | *37 | 29.3 | 40 |
| OpenChat | 7b | 71.3 (66.5) | 18 | 33.0 | *19 | 29.0 | *26 | 62.0 | 22 | 14.0 | *27 | 43.0 | 44 | 42.1 | 24 |
| Gemma-Inst | 7b | 28.0 (23.2) | *54 | 6.0 | *54 | 10.0 | *54 | 29.0 | 54 | 2.0 | *51 | 31.0 | 53 | 17.7 | 54 |
| Gemma | 7b | 31.7 (25.0) | 49 | 12.0 | *45 | 13.0 | *50 | 40.0 | *43 | 2.0 | *51 | 39.0 | *47 | 23.0 | 48 |
| | 2b | 22.0 (17.1) | 57 | 2.0 | 57 | 6.0 | 57 | 24.0 | 57 | 2.0 | *51 | 21.0 | 56 | 12.8 | 57 |
| Phi-2 | 2.7b | 50.0 (45.1) | *33 | 18.0 | *38 | 23.0 | *36 | 49.0 | *29 | 14.0 | *27 | 37.0 | 50 | 31.8 | 38 |
| Qwen-1.5 | 72b | 67.1 (61.6) | 23 | 28.0 | 24 | 28.0 | 28 | 61.0 | *23 | 21.0 | *17 | 47.0 | *33 | 42.0 | 25 |
| | 14b | 50.0 (45.7) | *33 | 20.0 | *36 | 23.0 | *36 | 48.0 | 31 | 18.0 | *21 | 44.0 | *39 | 33.8 | 31 |
| | 7b | 42.1 (37.8) | *41 | 16.0 | *41 | 13.0 | *50 | 43.0 | *37 | 7.0 | 39 | 32.0 | 52 | 25.5 | 45 |

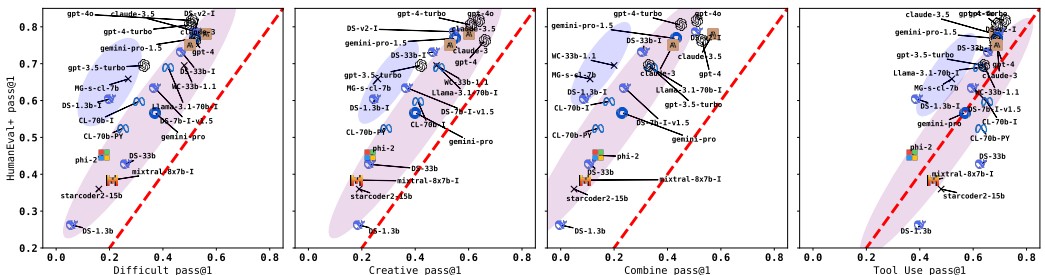

Figure 2: HUMANEVAL+ vs EVOEVAL pass@1. Red identity line shows equivalent performance. We cluster the LLMs into: purple region – aligned performance on HUMANEVAL vs. EVOEVAL and blue region – over performance on HUMANEVAL vs. EVOEVAL.

**LLMs struggle on EVOEVAL benchmarks compare to the high performance achieved on HUMANEVAL.** One surprising finding comes from SUBTLE, where the average performance of LLMs drops by 22.5% on the same 100 problems[2], even though only small changes are made to the original problems and the difficulty remains roughly the same. Appendix E Figure 21 presents an example problem and failing solution. Furthermore, we can also identify LLMs which struggle heavily on specific types of problems compared to their relative performance on HUMANEVAL. Figure 2 shows a scatter plot of HUMANEVAL+ vs. EVOEVAL scores. As we saw before, the significant portions of the models tend to be worse on EVOEVAL than HUMANEVAL (i.e., purple shaded region). However, there are LLMs that have a *much* higher HUMANEVAL score compared to their performance on EVOEVAL (i.e., blue shaded region). This implies potential data leakage of popular benchmarks where LLM performances are artificially inflated but do not translate to more difficult or other program synthesis problems.

**Significant ranking changes of LLMs on EVOEVAL.** Compared to HUMANEVAL where top models all perform similarly, we observe drastic differences in ranking changes on EVOEVAL. We observe that while the relative difference between the top 10 models on HUMANEVAL is around 10%, the difference on EVOEVAL on average is over 20%. Due to such saturation, existing benchmarks may not reliably rank the program synthesis ability of each model. For example, while Claude-3.5 and GPT-4-Turbo are tied for second on HUMANEVAL, they both excel at different types of problems: Claude-3.5 performs best on difficult and combine problems, while GPT-4-Turbo is better with tool use and creative tasks. Furthermore, while GPT-4o achieves the top HUMANEVAL and HUMANEVAL+ score, it falls off compared to the base GPT-4 variant where it is worse on DIFFICULT, CREATIVE and COMBINE problems. Such evaluation cannot be gained through naively reporting existing coding benchmark performance. Overall, by evolving the original benchmark into more difficult and diverse problems of different types, EVOEVAL can provide a more holistic evaluation and ranking of the coding ability of LLMs.

**EVOEVAL can be used to comprehensively compare multiple models.** In Figure 3, while both WizardCoder-1.1 and Phind-CodeLlama-2 have similar HUMANEVAL scores, they perform drastically differently across EVOEVAL benchmarks. WizardCoder-1.1 is better on DIFFICULT and CREATIVE while Phind-CodeLlama-2 is better on COMBINE problems. This can be explained through the training dataset used in each LLM: WizardCoder-1.1 uses an evolving dataset by generating more complex problems whereas Phind-CodeLlama-2 is fine-tuned on high quality programming problems that seems to boost the ability to solve programs which combines multiple programming concepts. Different from just reporting a singular pass@*k* score, EVOEVAL also allows a detailed analysis

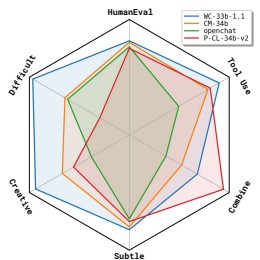

Figure 3: Radar graph

---

[2]Note that SUBTLE only contains 100 problems, and the pass@1 score on these 100 seed HUMANEVAL problems is higher compared to the full 164 problems. Therefore, this back-to-back performance drop is much higher than the performance drop from full HUMANEVAL to SUBTLE (5.0%) mentioned above.

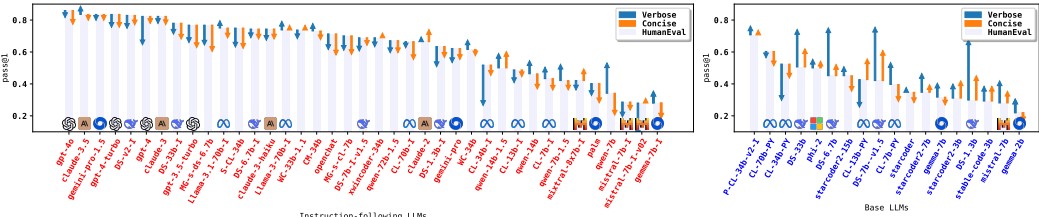

Figure 4: HUMANEVAL pass@1 with relative decrease or increase on VERBOSE and CONCISE.

Table 2: Results on COMBINE and COMBINE-NAIVE. HUMANEVAL is categorized into *pass both*, *one* and *none*, depending on the success on the two parent problems used for combination. COMBINE (Solved) and COMBINE-NAIVE (Solved) then show the distribution of solved problems that came from the previous categories. *Composition Percentage* is the % of *pass both* problems the LLM can *still* solve when combined.

| | Size | HUMANEVAL | | | COMBINE (Solved) | | | Composition Percentage |
|---|---|---|---|---|---|---|---|---|
| | | pass both | pass one | pass none | pass both | pass one | pass none | |
| GPT-4o | NA | 80 | 20 | 0 | 49 | 2 | 0 | **61.2%** |
| GPT-4-Turbo | NA | 79 | 19 | 2 | 38 | 6 | 1 | **48.1%** |
| GPT-4 | NA | 93 | 7 | 0 | 50 | 3 | 0 | **53.8%** |
| GPT-3.5-Turbo | NA | 65 | 34 | 1 | 24 | 9 | 0 | **36.9%** |
| Claude-3.5 | NA | 81 | 18 | 1 | 49 | 8 | 0 | **60.5%** |
| Claude-3 | NA | 81 | 19 | 0 | 35 | 7 | 0 | **43.2%** |
| Gemini-1.5-pro | NA | 86 | 13 | 1 | 40 | 3 | 0 | **46.5%** |
| DS Coder-v2-Inst | 236b | 83 | 16 | 1 | 47 | 4 | 0 | **56.6%** |

| | Size | HUMANEVAL | | | COMBINE-NAIVE (Solved) | | | |
|---|---|---|---|---|---|---|---|---|
| GPT-4o | NA | 881 | 185 | 8 | 589 | 50 | 0 | **66.9%** |
| GPT-4-Turbo | NA | 863 | 195 | 16 | 407 | 61 | 3 | **47.2%** |
| GPT-4 | NA | 1018 | 55 | 1 | 768 | 7 | 0 | **75.4%** |
| GPT-3.5-Turbo | NA | 799 | 261 | 14 | 474 | 79 | 1 | **59.3%** |
| Claude-3.5 | NA | 861 | 203 | 10 | 710 | 93 | 1 | **82.5%** |
| Claude-3 | NA | 796 | 268 | 10 | 359 | 96 | 1 | **45.1%** |
| Gemini-1.5-pro | NA | 788 | 267 | 19 | 595 | 148 | 6 | **75.5%** |
| DS Coder-v2-Inst | 236b | 805 | 252 | 17 | 598 | 95 | 5 | **74.3%** |

across different dimensions of coding capability to identify particular domains or types of coding questions an LLM struggles with or excels in.

**Instruction-following LLMs are sensitive to subtle changes or rephrasing in problem docstrings.** Figure 4 shows the HUMANEVAL score (bar) and the relative performance drop or improvement (arrows) on VERBOSE and CONCISE. We observe that almost all instruction-following LLMs drop in performance (average 3.4% and 4.0% decrease on VERBOSE and CONCISE respectively) when evaluated on the two semantic-preserving dataset compared to the original HUMANEVAL. This is drastically different from the base variants, where we even observe performance improvements (average 0.5% and 2.1% increase on VERBOSE and CONCISE respectively). VERBOSE and CONCISE do not change the semantic meaning of the original problem; they simply reword it in either a more verbose or concise manner. Prior work Deng et al. (2023a) has shown that by rephrasing the original problem description, one can further boost LLM performance, and we observe the similar phenomenon here mostly only for non-instruction-following models. Additionally, even on SUBTLE, where only small changes are applied, on average, instruction-following LLMs drops by 7.4% whereas base models only decrease by less than 1%. These findings across LLM types show that while instruction-tuned LLMs are expected to align better with detailed instructions, they fail to distinguish between these rephrasing or subtle changes in docstring, indicating potential memorization or contamination of prior evaluation benchmarks.

## 4.2 Problem Composition & Decomposition

**Composing problems.** The ability to compose different known concepts to solve new problems is known as *compositional generalization* (Keysers et al., 2020). This skill is essential for code synthesis, especially for complex problems in real-world programs. However, measuring compositional generalization in LLM presents a fundamental challenge since

Table 3: Results on DECOMPOSE. HUMANEVAL shows the pass/fail breakdown of the 50 seed HUMANEVAL problems. DECOMPOSE is categorized into *pass both*, *one* and *none*, based on if the LLM can solve both subproblems. *Decomp. %* and *Recomp. %* are the % of originally *passing* and *failing* problems for which the LLM can solve both subproblems respectively.

| | Size | HUMANEVAL | | DECOMPOSE | | | | | | Decomp. % | Recomp. % |
|---|---|---|---|---|---|---|---|---|---|---|---|
| | | | | HUMANEVAL pass | | | HUMANEVAL fail | | | | |
| | | pass | fail | pass both | pass one | pass none | pass both | pass one | pass none | | |
| 💲GPT-4o | NA | 41 | 9 | 26 | 14 | 1 | 5 | 4 | 0 | **63.4%** | **55.6%** |
| 💲GPT-4-Turbo | NA | 39 | 11 | 29 | 9 | 1 | 4 | 6 | 1 | **74.4%** | **36.4%** |
| 💲GPT-4 | NA | 47 | 3 | 37 | 10 | 0 | 0 | 3 | 0 | **78.7%** | **0.0%** |
| 💲GPT-3.5-Turbo | NA | 33 | 17 | 19 | 13 | 1 | 11 | 4 | 2 | **57.6%** | **64.7%** |
| Ⓐ Claude-3.5 | NA | 38 | 12 | 25 | 9 | 4 | 3 | 9 | 0 | **65.8%** | **25.0%** |
| Ⓐ Claude-3 | NA | 39 | 11 | 26 | 11 | 2 | 6 | 5 | 0 | **66.7%** | **54.5%** |
| Ⓞ Gemini-1.5-pro | NA | 41 | 9 | 27 | 13 | 1 | 5 | 3 | 1 | **65.9%** | **55.6%** |
| 🐋 DS Coder-v2-Inst | 236b | 38 | 12 | 31 | 7 | 0 | 6 | 6 | 0 | **81.6%** | **50.0%** |

it requires controlling the relationship between training and test distributions (Shi et al., 2024). While it is not easy to control the pre-training data of LLMs, we have more control in the testing phase. Hence, we focus on program concepts that have been demonstrated to fall within the capabilities of an LLM, and explore whether this proficiency extends to the combination of program concepts. As such, we start by taking a deeper look at the COMBINE problems evolved from combining previous HUMANEVAL problems.

First half of Table 2 shows the COMBINE dataset results of the top LLMs. We observe that almost all problems solved came from the pass both category, which is intuitive as we do not expect LLMs to solve a problem composed of subproblems that it cannot already solve. However, the composition percentage is quite low, as only a few LLMs are able to achieve greater than half. This demonstrates that while state-of-the-art LLMs can achieve a high pass rate on simple programming tasks, they still struggle with composing these known concepts to address more complex problems.

**Composing problems naively.** Since COMBINE problems are not guaranteed to contain no additional new concepts, we build a simplified dataset for sequential composition. Let $A$ and $B$ be two separate problems with $x$ as input(s) for $A$, we aim to create a new problem $C$ with the same inputs where the solution can be written as $B(A(x))$. To accomplish this, the new problem combines docstrings for A and B sequentially. However, simple concatenation of docstrings leads to unclear descriptions. As such, for each problem in HUMANEVAL, we manually create two separate variants based on which order the problem may come in the new docstring. Figure 5 shows an example of how naive combination problem is constructed with the manual sequential instruction highlighted in red.

```
def add(x: int, y: int):
    """add two numbers x and y"""
                    Problem A

def digits(n):
    """Given a positive integer n,
    return the product of the odd digits.
    Return 0 if all digits are even."""
                    Problem B

def add_digits(x: int, y: int):
    """First, add two numbers x and y

    Next, given the resulting
    positive integer n,
    return the product of the odd digits.
    Return 0 if all digits are even."""
                    Problem C
```

Figure 5: COMBINE-NAIVE problem

Using these modified problem docstrings, we build a sequential combination dataset – COMBINE-NAIVE, containing 1074 problems by randomly combining problems filtering for input output matching (i.e., the type of $A(x)$ should equal to the type of $y$ in $B(y)$).

The latter half of Table 2 shows the results on COMBINE-NAIVE following the same setup as COMBINE. We observe that while the composition percentage on the naive dataset improves significantly compared to the evolved COMBINE dataset, it still fails to reach near perfection, with the best LLM being able to only solve ∼80% of prior pass both problems. While existing LLM training or inference paradigms for code focus on obtaining high quality datasets boosted with instruction-tuning, our result shows that existing LLMs still struggle with the concept of problem composition to tackle more complex problems.

**Decomposing problems.** We also evaluate *problem decomposition* – decomposing larger problems into multiple subproblems. We start by selecting 50 HUMANEVAL problems and then follow our approach in Section 2 to decompose each original problem into two smaller subproblems, creating 100 problems in our DECOMPOSE benchmark. Table 3 shows the

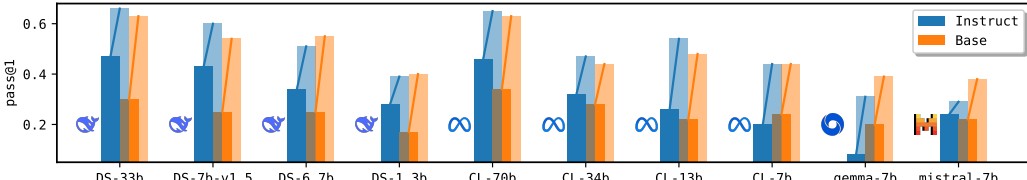

Figure 6: pass@1 from TOOL_USE-MAIN_ONLY (darker bar) to TOOL_USE (lighter bar).

results of selected LLMs on DECOMPOSE. We first observe that similar to the composition percentage in the COMBINE and COMBINE-NAIVE problems, LLMs do not achieve a high decomposition percentage. Since current LLMs are trained to recover seen outputs in their training data, and when used for program synthesis, they cannot generalize the concepts from training data. This is demonstrated by not being able to solve smaller subproblems obtained from solved more difficult parent problems. Conversely, LLMs can sometimes solve both subproblems even when the parent problem is not solved (i.e., recomposition percentage), showing room for improvement with techniques like planning (Jiang et al., 2023b) and least-to-most prompting (Zhou et al., 2022).

### 4.3 Tool Use Problems

We analyze the TOOL_USE benchmark, which contains helper functions. We further construct the TOOL_USE-MAIN_ONLY benchmark, which contains the same set of problem as TOOL_USE, except that the input to the LLM does not include any helpers. We observe that compared to without any helper functions (average 29.8%), LLMs on average improve by 80.1% when provided with helper functions. This is expected as helper functions reduce the work required to solve the more complex problem. However, this improvement is not uniform: the average improvement when given the auxiliary functions for instruction-following models is only 59.2% compared to the base LLMs' improvement of 122.0%.

In Figure 6, we observe that without the helpers, the instruction-following models significantly outperform their base LLMs. However, once the helpers are provided, this gap is drastically decreased, with cases even where the base models outperform their instruction-following counterparts. As real-world coding involves understanding, using, and then reusing existing functions across different places in the repository, being able to successfully leverage auxiliary methods is key. Current instruction-following LLMs are generally fine-tuned with data consisting of self-contained code snippets without the interaction and learning of function usages. This is further exacerbated by prior benchmarks, which mostly use self-contained functions, thus cannot test the tool-using capability of LLMs.

## 5 Related Work

**Large language models for code.** Starting with the general development of LLMs for general purpose tasks, developers have applied LLMs to perform code-related tasks by further training LLMs using code snippets from open-source repositories. Such LLMs include CODEX (Chen et al., 2021), CodeT5 (Wang et al., 2021), CodeGen (Nijkamp et al., 2023), InCoder (Fried et al., 2023), CodeLlama (Rozière et al., 2023), StarCoder (Li et al., 2023; Lozhkov et al., 2024), DeepSeek Coder (Guo et al., 2024), etc. More recently, researchers have applied instruction-tuning methods to train code-specific LLMs that are well-versed in following instructions. Examples of such LLMs include CodeLlama-Inst (Rozière et al., 2023) and DeepSeek Coder-Instruct (Guo et al., 2024). WizardCoder (Luo et al., 2023) instruction-tunes the model using Evol-Instruct to create more complex instructions. Magicoder (Wei et al., 2023) develops OSS-Instruct by synthesizing high quality instruction data from open-source code snippets. OpenCodeInterpreter (Zheng et al., 2024) leverages execution feedback for instruction-tuning in order to better support multi-turn code generation and refinement.

**Program synthesis benchmarking.** HUMANEVAL (Chen et al., 2021) and MBPP (Austin et al., 2021) are two of the most widely-used handcrafted code generation benchmarks complete with test cases. Building on these popular benchmarks, additional variants have been

crafted including: EVALPLUS (Liu et al., 2023) which improves the two benchmarks with more complete test cases; HUMANEVAL-X (Zheng et al., 2023) which extends HUMANEVAL to C++, JavaScript and Go; MultiPL-E (Cassano et al., 2023) which further extends both HUMANEVAL and MBPP to 18 languages. Similarly, other benchmarks have been developed for specific domains: DS-1000 (Lai et al., 2023) and Arcade (Yin et al., 2022) for data science APIs; CodeContests (Li et al., 2022), APPS (Hendrycks et al., 2021), and LiveCodeBench (Jain et al., 2024) for programming contests, and SWE-Bench (Jimenez et al., 2024) for software engineering tasks. Different from prior benchmarks which require handcraft problems from scratch – high manual effort or scrape open-source repositories or coding contest websites – leading to unavoidable data leakage, EVOEVAL directly uses LLMs to *evolve* existing benchmark problems to create new complex evaluation problems. Furthermore, contrasting with the narrow scope of prior benchmarks (often focusing on a single type or problem, i.e., coding contests), EVOEVAL utilizes targeted transformation to evolve problems into different domains, allowing for a more holistic evaluation of program synthesis using LLMs.

## 6 Conclusion

We present EVOEVAL– a set of program synthesis benchmarks created by *evolving* existing problems into different target domains for a holistic and comprehensive evaluation of LLM program synthesis ability. Our results on **57** LLMs show drastic drops in performance (average 38.1%) when evaluated on EVOEVAL. Additionally, we observe significant ranking differences compared to prior leaderboards, indicating potential dataset overfitting on existing benchmarks. We provide additional insights, including the brittleness of instruction-following LLMs as well as the limited compositional generalization abilities of LLMs.

## 7 Acknowledgment

We thank Owen Colegrove for his help on starting this project and providing valuable feedback throughout, Jiawei Liu for providing helpful discussions and Yifeng Ding for his help in running experiments. This work was partially supported by NSF grant CCF-2131943 and Kwai Inc. This project is supported, in part, by funding from Two Sigma Investments, LP. Any opinions, findings, and conclusions or recommendations expressed in this material are those of the authors and do not necessarily reflect the views of Two Sigma Investments, LP.

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

## A    Reproducibility Statement

We provide the detailed setup and steps to reproduce our results in the README file of our supplementary material. Furthermore, we have prepared a public repository, package, and dataset release to support open-source usage (these are not linked here to preserve anonymity). Here we describe our computation setup and estimated runtime to reproduce the full experimental result of EVOEVAL.

**Compute.** We perform all our evaluation on an Ubuntu 22.04 linux machine with 96 cores (192 threads), 512GB RAM, 8 NVIDIA RTX A6000 GPUs with 50GB VRAM each. For LLMs that can only be accessible via endpoints, we use the provided commercial endpoints and do not require local GPU resources for those models.

**Time.** Due to the cost of running LLMs, to completely regenerate all 57 LLMs' solutions for all benchmarks we estimate it will take several days depending on available GPU resource. However, we provided the raw LLM samples generated by us in the supplementary material. To perform evaluation on the LLM samples and regenerate our reported results, we estimate it will take around 1 hour to complete.

## B    EVOEVAL Benchmarks

Table 4: EVOEVAL and HUMANEVAL benchmark statistics. Note: the number in bracket shows the number of testcases in the augmented HUMANEVAL+ benchmarks, in EVOEVAL, they are directly reused in SUBTLE, VERBOSE and CONCISE due the similarity.

| | original | semantic-altering | | | | | semantic-preserving | |
| --- | --- | --- | --- | --- | --- | --- | --- | --- |
| | HUMANEVAL | DIFFICULT | CREATIVE | SUBTLE | COMBINE | TOOL_USE | VERBOSE | CONCISE |
| # of problems | 164 | 100 | 100 | 100 | 100 | 100 | 164 | 164 |
| Avg. problem len. | 450.6 | 749.4 | 982.1 | 406.8 | 860.4 | 1224.6 | 450.6 | 450.6 |
| Avg. # test cases | 9.6 (764.1) | 49.8 | 43.1 | 10.3 (745.4) | 51.8 | 51.3 | 9.6 (764.1) | 9.6 (764.1) |

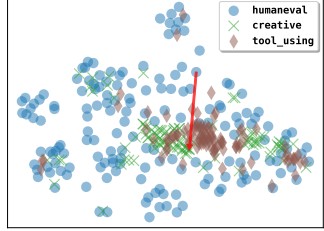 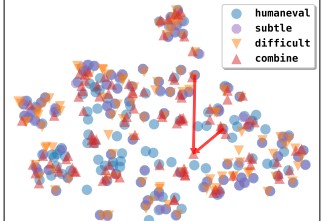 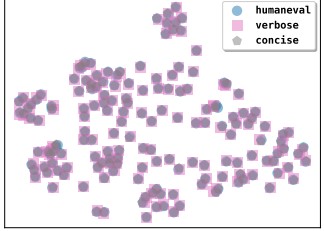

   (a) CREATIVE & TOOL_USE    (b) SUBTLE,DIFFICULT,COMBINE    (c) VERBOSE & CONCISE

Figure 7: 2 dimensional t-SNE visualization of EVOEVAL benchmarks.

Problems in EVOEVAL consist mainly of self-contained functions, except for TOOL_USE that includes helper functions specifically designed to test the tool using capability of LLMs. Each problem uses a docstring to illustrate the problem specification, along with test cases and ground truth to evaluate the functional correctness. Table 4 shows the statistics of the benchmarks in EVOEVAL. In total, EVOEVAL includes 828 problems across 7 different datasets (5 semantic-altering and 2 semantic-preserving).

Figure 7 shows the embedding visualization using t-SNE (Hinton & Roweis, 2002) by projecting high-dimension representation of the problems docstrings in both EVOEVAL and HUMANEVAL into the 2D plane. We produce this embedding using text-embedding-3-large model from OpenAI with t-SNE perplexity=50 and iter=1000. First, we see that CREATIVE and TOOL_USE drastically change the embedding distribution compared to the original dataset. The arrow in Figure 7a shows one example of the shift in distribution from the original problem to a creative one. Next, we see that SUBTLE, DIFFICULT and COMBINE largely retain the same distribution as the original problems. This is due to the high parity across these problem descriptions where SUBTLE only applies subtle changes and DIFFICULT adds additional complex constraints while keeping the main problem descriptions largely the same. Specifically, for COMBINE, we can see from an example arrow in Figure 7b, the

new combined problem shifts the embedding for both of the original problems. Finally, we observe that for VERBOSE and CONCISE, the embeddings almost perfectly match the original problem, reflecting their semantic-preserving nature. In Appendix E, we present example problems for each benchmark in EVOEVAL.

### B.1 Potential Bias in Performance

Using GPT-4 to transform the original HUMANEVAL problems into new domains may introduce potential biases. For instance, LLMs trained on data generated by GPT-4 might unfairly benefit from our benchmark, as their training data could be similar to the transformed problems. We note here that we manually examine all problems transformed by GPT-4 (including each ground truth solution) and adjust any faults or incorrect solutions, as such, the problem descriptions are not all purely generated by GPT-4. Moreover, our method of evolving existing problems to new domains is similar to existing instruction-tuning methods (e.g., Evo-Instruct (Luo et al., 2023)), which may also favor these LLMs trained with similar evolved datasets. Another potential bias also comes from the usage of GPT-4, as it might create problems that reflect its own bias or patterns, resulting in a lack of diversity. In future work, we hope to apply additional state-of-the-art LLMs to perform our transformation to improve problem diversity.

## C  Evaluation LLMs

### C.1 Evaluated LLMs

Table 5 and  6 shows the overview of the 57 LLMs we evaluated in our work. For any LLMs that provide their open-source weights, we directly obtain them from HuggingFace model hub (HuggingFace)[3]. For any close-sourced LLMs, we directly access their model endpoints using their providers.

### C.2  Detailed Evaluation Setup

**LLM generation.** As mentioned in Section 3, we report the pass@1 score for each LLM on our dataset generated using greedy decoding (i.e., sampling with temperature = 0). For each LLM, we provide a specific input prompt depending on the model type. For base LLMs (i.e., not instruction-following variants), we use only the function headers as input. For instruction-following, we make the best effort to follow examples provided by each model maker on the exact instruction and format to use at the time of writing. Specifically, for instruction-following LLMs, we ask the model to return the code snippet wrapped by code blocks (i.e., ```). Figure 8 shows an example input for GPT-4 on a CREATIVE problem.

Furthermore, we also provide a custom sanitization script adopted from EVALPLUS (Liu et al., 2023) which parses the raw LLM outputs for code block parsing (e.g., removing ``` indicators for instruction-following models) and end-of-string identifiers (e.g., removing tokens like ). Each model generated output is passed into the sanitization script and the evaluation occurs on the sanitized outputs.

**Oracle.** To evaluate the functional correctness of each LLM synthesized solution, we use differential testing by comparing the model output with the ground truth output on a set of test inputs. We build our evaluation framework on top of the EVALPLUS evaluation script used for HUMANEVAL and HUMANEVAL+ benchmark which evaluates multiple problems and solutions in parallel for efficiency. For each test case, we perform exact matching or check if the output is within an absolute difference threshold of $10^{-6}$ if the output is a floating point type. We additionally implement our evaluation script by recursively checking the type and performing the appropriate comparison (e.g., dictionary outputs are first length checked for equivalence and then matching is done for each value and key). Furthermore, we also implement custom oracles for specific problems where there could be multiple solutions or

---

[3]For certain LLMs, we may use the vLLM inference library for more efficient generation

Table 5: Detailed overview of evaluated models. Model ID indicates either the API endpoint name or huggingface model name used for the particular model. Available Weights indicate whether the model is evaluated by accessing a close-sourced API endpoint or ran locally with provided weights. Note: 💬 denotes instruction-following LLMs

| | Size | Model ID | Available Weights |
|---|---|---|---|
| GPT-4o💬(OpenAI, 2024) | NA | gpt-4o-2024-05-13 | ✗ |
| GPT-4-Turbo💬(OpenAI, 2023) | NA | gpt-4-0125-preview | ✗ |
| GPT-4💬(OpenAI, 2023) | NA | gpt-4-0613 | ✗ |
| GPT-3.5-Turbo💬(OpenAI, 2022) | NA | gpt-3.5-turbo-0125 | ✗ |
| Claude-3.5💬(Anthropic, 2024b) | NA | claude-3-5-sonnet-20240620 | ✗ |
| Claude-3💬(Anthropic, 2024a) | NA | claude-3-opus-20240229 | ✗ |
| Claude-3-haiku💬(Anthropic, 2024a) | NA | claude-3-haiku-20240307 | ✗ |
| Claude-2💬(Anthropic, 2023) | NA | claude-2.1 | ✗ |
| Gemini-1.5-pro💬(Google, 2024) | NA | gemini-1.5-pro | ✗ |
| Gemini💬(Team et al., 2023) | NA | gemini-1.0-pro | ✗ |
| PaLM-2💬(Anil et al., 2023) | NA | text-bison-001 | ✗ |
| DS Coder-v2-Inst💬(Zhu et al., 2024) | 236b | DeepSeek-Coder-V2-0724 | ✗ |
| DS Coder-Inst💬(Guo et al., 2024) | 33b
6.7b
1.3b | deepseek-ai/deepseek-coder-33b-instruct
deepseek-ai/deepseek-coder-6.7b-instruct
deepseek-ai/deepseek-coder-1.3b-instruct | ✓
✓
✓ |
| DS Coder (Guo et al., 2024) | 33b
6.7b
1.3b | deepseek-ai/deepseek-coder-33b-base
deepseek-ai/deepseek-coder-6.7b-base
deepseek-ai/deepseek-coder-1.3b-base | ✓
✓
✓ |
| DS Coder-1.5-Inst.💬(Guo et al., 2024) | 7b | deepseek-ai/deepseek-coder-7b-instruct-v1.5 | ✓ |
| DS Coder-1.5 (Guo et al., 2024) | 7b | deepseek-ai/deepseek-coder-7b-base-v1.5 | ✓ |
| Llama -3.1-Inst💬(Meta, 2024) | 70b | meta-llama/Meta-Llama-3.1-70B-Instruct | ✓ |
| Llama -3-Inst💬(Dubey et al., 2024) | 70b | meta-llama/Meta-Llama-3-70B-Instruct | ✓ |
| CodeLlama-Inst💬(Rozière et al., 2023) | 70b
34b
13b
7b | codellama/CodeLlama-70b-Instruct-hf
codellama/CodeLlama-34b-Instruct-hf
codellama/CodeLlama-13b-Instruct-hf
codellama/CodeLlama-7b-Instruct-hf | ✓
✓
✓
✓ |
| CodeLlama (Rozière et al., 2023) | 70b
34b
13b
7b | codellama/CodeLlama-70b-Python-hf
codellama/CodeLlama-34b-Python-hf
codellama/CodeLlama-13b-Python-hf
codellama/CodeLlama-7b-Python-hf | ✓
✓
✓
✓ |
| WizardCoder💬(Luo et al., 2023) | 34b | WizardLM/WizardCoder-Python-34B-V1.0 | ✓ |
| WizardCoder-1.1💬(Luo et al., 2023) | 33b | WizardLM/WizardCoder-33B-V1.1 | ✓ |
| XwinCoder💬(Team, 2023) | 34b | Xwin-LM/XwinCoder-34B | ✓ |
| Phind-CodeLlama-2 (phind team, 2023) | 34b | Phind/Phind-CodeLlama-34B-v2 | ✓ |
| Code Millenials💬(BudEcosystem) | 34b | budecosystem/code-millenials-34b | ✓ |
| Speechless-CL💬(Su) | 34b | uukuguy/speechless-codellama-34b-v2.0 | ✓ |
| Magicoder-s-DS💬(Wei et al., 2023) | 6.7b | ise-uiuc/Magicoder-S-DS-6.7B | ✓ |
| Magicoder-s-CL💬(Wei et al., 2023) | 7b | ise-uiuc/Magicoder-S-CL-7B | ✓ |
| StarCoder2 (Lozhkov et al., 2024) | 15b
7b
3b | bigcode/starcoder2-15b
bigcode/starcoder2-7b
bigcode/starcoder2-3b | ✓
✓
✓ |
| StarCoder (Li et al., 2023) | 15b | bigcode/starcoder | ✓ |
| Mixtral-Inst💬(Mistral AI team, 2023) | 8x7b | mistralai/Mixtral-8x7B-Instruct-v0.1 | ✓ |
| Mistral-Inst-v02💬(Jiang et al., 2023a) | 7b | mistralai/Mistral-7B-Instruct-v0.2 | ✓ |
| Mistral-Inst💬(Jiang et al., 2023a) | 7b | mistralai/Mistral-7B-Instruct-v0.1 | ✓ |
| Mistral (Jiang et al., 2023a) | 7b | mistralai/Mistral-7B-v0.1 | ✓ |
| OpenChat💬(Wang et al., 2023) | 7b | openchat/openchat-3.5-0106 | ✓ |
| stable-code (Pinnaparaju et al.) | 3b | stabilityai/stable-code-3b | ✓ |
| Gemma-Inst.💬 (Team et al., 2024) | 7b | google/gemma-7b-it | ✓ |
| Gemma (Team et al., 2024) | 7b
2b | google/gemma-7b
google/gemma-2b | ✓
✓ |

Table 6: Detailed overview of evaluated models continued. Model ID indicates either the API endpoint name or huggingface model name used for the particular model. Available Weights indicate whether the model is evaluated by accessing a close-sourced API endpoint or ran locally with provided weights. Note: 💬 denotes instruction-following LLMs

| | Size | Model ID | Available Weights |
|---|---|---|---|
| 🪟 Phi-2 (Microsoft Research, 2023) | 2.7b | microsoft/phi-2 | ✓ |
| | 72b | Qwen/Qwen1.5-72B-Chat | ✓ |
| Qwen-1.5💬(Bai et al., 2023b) | 14b | Qwen/Qwen1.5-14B-Chat | ✓ |
| | 7b | Qwen/Qwen1.5-7B-Chat | ✓ |
| Qwen💬(Bai et al., 2023a) | 14b | Qwen/Qwen-14B-Chat | ✓ |
| | 7b | Qwen/Qwen-7B-Chat | ✓ |

```
Please complete the following code snippet.
```
def transform_canvas(canvas: str) -> str:
    """
    You have an canvas containing either '#' (representing a wall), '-' (
    ↪ representing
    an empty space), or 'P' (representing the point at which a painter starts). The
    ↪ painter
    can move horizontally on the canvas and paints all empty spaces he encounters
    with '*' without crossing or hitting the walls.

    The task is to return an updated canvas with all the accessible spaces painted,
    keeping wall configuration and unaccessible spaces same. If the canvas contains
    ↪ no painter 'P',
    return the canvas as it is. If there are more than one 'P' or the number of
    ↪ painted space divides the empty spaces evenly, return 'Invalid canvas'.

    Examples:

    >>> transform_canvas('P----#-----#-----#-----')
    'P****#-----#-----#-----'

    >>> transform_canvas('--#-P#-----#-----#--#--')
    'Invalid canvas'

    >>> transform_canvas('-----#--P--#-----#-----')
    '-----#**P**#-----#-----'

    >>> transform_canvas('-----#-----#--P---#P----')
    'Invalid canvas'
    """
```
```

Figure 8: Example input prompt for GPT-4

simple tolerance or exact matching cannot fully guarantee correctness. Additionally, we also use timeout as another evaluation method. Our setting again follows EVALPLUS default setup where the timeout per problem is defined as $T = max(T_{max}, f \times t_{gt})$ with default values of $T_{max} = 1000ms$, $f = 4$ and $t_{gt}$ defined as the measured ground truth solution time to produce the correct output. All timeout related factors can be adjusted to account for variance on different underlying machine and hardware.

## D Transformation Prompts

Here we provide the exact targeted transformation prompts used to evolve existing benchmark problems. Figure 9, 10, 11, 12, 13, 14, 15, and 16 show the prompt for DIFFICULT, CREATIVE, SUBTLE, COMBINE, TOOL_USE, VERBOSE, CONCISE and DECOMPOSE respectively.

```
Here is an example coding problem:
```
{problem}
```

Please increase the difficulty of the given coding problem

You can increase the difficulty using the following method:
- Add new constraints and requirements to the original problem, adding
    ↪ approximately 10 additional words.
- Replace a commonly used requirement in the programming task with a less common and
    ↪  more specific one.
- Add more reasoning steps.

Return the new problem in the same format as the example problem (i.e., ```function
    ↪ header + docstring```)
```

Figure 9: Prompt for DIFFICULT

```
Here is an example coding problem:
```
{problem}
```

Please generate a more creative coding problem.
You should avoid common programming concepts and instead focus on creating a
    ↪ problem that is interesting and fun to solve.
Return the new problem in the same format as the example problem (i.e., ```function
    ↪ header + docstring```)
```

Figure 10: Prompt for CREATIVE

```
Please add a subtle and simple change to the given problem.

You can change the problem using, but not limited to, the following methods:

Add one new requirement to the original problem, such as "Return the list in
    ↪ ascending order", "Return the list in ascending alphabetical order" and "
    ↪ Return unique elements only".

Invert one requirement of the original problem; for instance, reverse the
    ↪ instruction "from shortest to longest" to "from longest to shortest",
    ↪ reverse "maximum" to "minimum", or reverse "the first" to "the last".

Replace one requirement with another similar but different one; for example, if the
    ↪  original problem requires the values to be sorted, change it to keeping the
    ↪  original order.

Replace constants; for instance, replace zero with one.

Please only apply a minor change, ensuring that the new problem remains logical.
    ↪ Return the new problem in the same format as the original problem (```...
    ↪ ```)

Below is the question:
```
{problem}
```
```

Figure 11: Prompt for SUBTLE

```
Here are two example problems:

Example problem 1:
```
{problem_1}
```

Example problem 2:
```
{problem_2}
```

Please create a new problem that combines problem 1 with problem 2 in a logical way.
    ↪  The new problem should seamlessly integrate the concepts from the two
    ↪ previous examples into a novel context, and require a solution that
    ↪ exercises the understanding of the concepts from both problems. It does not
    ↪ need to take all the inputs from both problems.

An incorrect way would be for the new problem to simply return the answers of the
    ↪ two problems separately. Another incorrect method would be to pass all
    ↪ inputs from both problems but some of them are not used to compute the
    ↪ output.

Return the new problem in the same format as the example problems in ```...```. Keep
    ↪  the docstring concise, and ensure the problem remains simple and clear.
```

Figure 12: Prompt for COMBINE

```
Here is an example coding problem:
```
{problem}
```

Please come up with a new problem which uses helper functions to solve the problem.

The new problem should contain the following:
first: one or more helper functions
second: the main problem description consist of the function header and docstring

The main problem description should not refer to the helper function(s) in any way.

The helper function(s) should implement simple parsing or checking logic.

To solve the main problem, one should also use additional complex logic than just
    ↪ calling the helper function(s).

Avoid problems on simple math concepts such as prime, palindrome, anagrams,
    ↪ factorial

Avoid concepts like emails, string or parsing-based problems

Please return the full implementation of the helper function(s) and the main
    ↪ problem description (not the implementation) in the same format as the
    ↪ example problem (```...```)
```

Figure 13: Prompt for TOOL_USE

Figure 17 and 18 show the refinement and I/O extraction/fixing prompt used in EVOEVAL. The refinement prompt is used to refine the original generated problem when inconsistency is detected (see Section 2). The extraction prompt is used to initially obtain a set of testcases from the problem docstring used for self-consistency evaluation. We further use an I/O fixing prompt (also in Figure 18) to fix any examples in the docstring which do not contain the right output (as computed by the ground truth generated by GPT-4).

```
Below is a coding problem

```
{problem}
```

Make the docstring more verbose and detailed but preserve the semantic meaning
Ensure the function name, input argument names, and example input/output are the
    ↪ same
Return the transformed problem in the same format as the original problem (i.e.,
    ↪ function header + docstring)
```

Figure 14: Prompt for VERBOSE

```
Below is a coding problem

```
{problem}
```

Make the docstring shorter and more concise but preserve the semantic meaning
Ensure the function name, input argument names, and example input/output are the
    ↪ same
Return the transformed problem in the same format as the original problem (i.e.,
    ↪ function header + docstring)
```

Figure 15: Prompt for VERBOSE

```
Below is a complex coding problem
```
{problem}
```

Please decompose the above into 2 smaller sub problems
Return the two modified problems in the same format as the initial problem (i.e.,
    ↪ ```function header + docstring```)
```

Figure 16: Prompt for DECOMPOSE

```
Below is a coding problem
```
{problem}
```

Ensure logical coherence in the given problem.
Improve the docstring's clarity and conciseness.
Fix missing or helpful imports.
Include example input/output if absent.
Return the modified problem in the same format as the example problem (i.e.,
    ↪ ```function header + docstring```)
```

Figure 17: Refinement prompt

# E  Example Problems in EVOEVAL

Here we demonstrate a few example problems across the benchmarks in EVOEVAL and corresponding GPT-4 solution which cannot solve the problem. Figure 19, 20, 21, 22, 23, 24, and 25 show such example for the EVOEVAL DIFFICULT, CREATIVE, SUBTLE, COMBINE, TOOL_USE, VERBOSE and CONCISE respectively.

```
Here is a function header with docstring:
```
{problem}
```
Please extract the example raw input argument and expected output from the
    ↪ docstring.
If there are no example input and output please provide new ones.
Return each pair of input and output as assertions in this format:
```
assert {function_name}({{the_first_input_example}}) == {{the_first_output_example
    ↪ }}
assert {function_name}({{the_second_input_example}}) == {{
    ↪ the_second_output_example}}
...
```

Here is a problem with docstring:
```
{problem}
```
Some example inputs and outputs in the docstring may be wrong. Please correct them
    ↪ according to the provided correct assertions below, and ensure that the
    ↪ correct example inputs and outputs ```
{assertions}
```
Return the revised problem in the same format as the original problem (i.e., ```any
    ↪ helper functions + function header + docstring```).
```

Figure 18: Input extraction and fixing prompts

```python
def common(l1: list, l2: list, n: int):
    """
    Return the n longest strings, sorted by increasing length that are common in two
    ↪  lists.
    However, in the case of a tie, prioritize the string that appears first in list1.
    ↪
    >>> common(["apple", "banana", "cherry", "date", "elderberry"], ["banana", "
    ↪ date", "fig", "grape", "elderberry"], 3)
    ["banana", "date", "elderberry"]
    >>> common(["cat", "dog", "elephant", "fox", "goat"], ["dog", "fox", "cat", "
    ↪ horse", "iguana", "goat"], 3)
    ["cat", "dog", "goat"]
    >>> common(["hello", "goodbye", "yes", "no", "maybe", "absolutely", "never"], ["
    ↪ no", "maybe", "absolutely", "always", "sometimes"], 1)
    ["absolutely"]
    """
    common_strings = [string for string in l1 if string in l2]
    common_strings.sort(key=len) # does not adjust for tie sorting scenario.
    return common_strings[-n:]
```

Figure 19: GPT-4 failing solution on problem EVOEVAL/58 in DIFFICULT

```python
def crossword_validator(grid: list, words: list):
    """
    Consider a 2D grid representing a square crossword puzzle. Each cell of the grid
     ↪   contains a single character.

    Determine if all the words from a provided list of words can be found in the grid.
     ↪
    Words can be found left-to-right (in a row), right-to-left (in a row),
    top-to-bottom down (in a column), and bottom-to-top (in a column).

    We assume that words are only found in contiguous cells and words do not 'wrap'
     ↪ around the grid.

    Note: the grid is a square (N x N) and all words are lowercase.

    Parameters:
    grid: list of list of char - where each char is a lowercase letter.
    words: list of str - where each str is a lowercase word.

    Returns:
    boolean - return True all words are found in the grid, False otherwise.

    >>> crossword_validator([['c', 'a', 't'], ['d', 'o', 'g'], ['d', 'o', 'p']], ['
     ↪ cat', 'dog', 'pop'])
    False
    >>> crossword_validator([['c', 'a', 't'], ['d', 'o', 'g'], ['d', 'o', 'p']], ['
     ↪ cdd', 'dog', 'pod'])
    True
    """

    n = len(grid)
    for word in words:
        found = False
        for i in range(n):
            # does not correctly check if a word can be found as a sublist of column/
     ↪ grid
            if ''.join(grid[i]) == word or ''.join(grid[i][::-1]) == word:
                found = True
                break
            if ''.join([grid[j][i] for j in range(n)]) == word or ''.join([grid[j][i]
     ↪  for j in range(n)][::-1]) == word:
                found = True
                break
        if not found:
            return False
    return True
```

Figure 20: GPT-4 failing solution on problem EVOEVAL/21 in CREATIVE

```python
def vowels_count(s):
    """Write a function vowels_count which takes a string representing
    a word as input and returns the number of vowels in the string.
    Vowels in this case are 'a', 'e', 'i', 'o', 'u'. Here, 'y' is also a
    vowel, but only when it is at the end of the given word.
    Only count the lowercase vowels.

    Example:
    >>> vowels_count("abcde")
    2
    >>> vowels_count("YACED")
    0
    """
    vowels = 'aeiou'
    count = 0
    for char in s:
        if char in vowels:
            count += 1
    # miss empty string case leading to crash
    if s[-1] == 'y':
        count += 1
    return count
```

Figure 21: GPT-4 failing solution on problem EVOEVAL/64 in SUBTLE

```python
def grade_with_same_chars(grades: list, s0: str, s1: str):
    """
    A teacher has a special grading system where she assigns her students letter
    ↪ grades
    based on their GPA according to the following table:

            GPA     |   Letter grade
            4.0             A+
          > 3.7             A
          > 3.3             A-
          > 3.0             B+
          > 2.7             B
          > 2.3             B-
          > 2.0             C+
          > 1.7             C
          > 1.3             C-
          > 1.0             D+
          > 0.7             D
          > 0.0             D-
            0.0             E

    However, this year she adds an extra condition: the students can only receive a
    ↪ grade
    if their name (s0) and the course title (s1) have the same characters. The
    ↪ function
    receives a list of GPAs and two strings (s0 and s1) and returns a list of grades
    for the students that pass the condition, or an empty list if no student passes.

    Example:
    grade_with_same_chars([4.0, 3, 1.7, 2, 3.5], 'abcd', 'dcba') ==> ['A+', 'B', 'C
    ↪ -', 'C', 'A-']
    grade_with_same_chars([4.0, 3, 1.7, 2, 3.5], 'aaabcd', 'dddcba') ==> ['A+', 'B',
    ↪  'C-', 'C', 'A-']
    grade_with_same_chars([4.0, 3, 1.7, 2, 3.5], 'abcd', 'dcbae') ==> []
    """
    # does not correctly check characters are same
    if sorted(s0) != sorted(s1):
        return []
    grading_system = {4.0: 'A+', 3.7: 'A', 3.3: 'A-', 3.0: 'B+', 2.7: 'B', 2.3: 'B-',
    ↪  2.0: 'C+', 1.7: 'C', 1.3: 'C-', 1.0: 'D+', 0.7: 'D', 0.0: 'D-', 0.0: 'E'}
    grades_assigned = []
    for gpa in grades:
        for key in sorted(grading_system.keys(), reverse=True):
            if gpa >= key:
                grades_assigned.append(grading_system[key])
                break
    return grades_assigned
```

Figure 22: GPT-4 failing solution on problem EVOEVAL/28 in COMBINE

```python
def is_valid_passport(passport: str) -> bool:
    """
    Checks whether a given string is a valid passport number.
    A valid passport number is defined as having exactly 9 alphanumeric characters.

    >>> is_valid_passport('P12345678')
    True
    >>> is_valid_passport('P123456789')
    False
    >>> is_valid_passport('P12345G78')
    False
    """

    if len(passport) != 9:
        return False
    if not passport.isalnum():
        return False
    return True

from typing import List, Tuple

def process_passport_data(data: str) -> List[Tuple[str, str]]:
    """
    Parse a string containing passport data and return a list of tuples.
    Each tuple contains the passport number and the name of the passport holder.
    The passport data string is in the format of 'PassportNumber:Name'
    separated by lines.

    Only process valid passport data. A valid passport number is defined as having
     ↪ exactly 9 alphanumeric characters.

    >>> process_passport_data('P12345678:John Doe\nX12345678:Jane Doe')
    [('P12345678', 'John Doe')]
    >>> process_passport_data('P12345678:John Doe\nA123456789:Jane Doe')
    [('P12345678', 'John Doe')]
    """
    valid_passports = []
    lines = data.split('\n')
    for line in lines:
        (passport, name) = line.split(':') # directly splits without checking for
     ↪ validity
        if is_valid_passport(passport):
            valid_passports.append((passport, name))
    return valid_passports
```

Figure 23: GPT-4 failing solution on problem EVOEVAL/23 in TOOL_USE

```python
def starts_one_ends(n):
    """
    This function, given a user-provided positive integer 'n', calculates and
     ↪ returns the total count of n-digit positive integers where 1 appears either
     ↪ at the start or the end of the number.

    For instance, if the input is 2, the function will count all the two-digit
     ↪ numbers that either start or end with 1, like 10, 11, 21, 31, ..., 91, and
     ↪ 12, 13, ..., 19.

    It is important to note that this function only accepts positive integers. If a
     ↪ non-integer or non-positive integer is provided, the function may not
     ↪ behave as expected.

    Args:
    n : int
        A positive integer that represents the number of digits in the numbers to be
     ↪ counted.

    Returns:
    int
        The count of the numbers of n-digit positive integers that start or end with
     ↪ 1.

    Examples:
    The function can be called in the following way:
        starts_one_ends(2)
    This example call would return the count of two-digit numbers where 1 appears at
     ↪  the start or at the end.
    """
    if n == 1:
        return 1
    else:
        # incorrectly adds 2 to the final answer
        return 18 * 10 ** (n - 2) + 2
```

Figure 24: GPT-4 failing solution on problem HUMANEVAL/83 in VERBOSE

```python
def circular_shift(x, shift):
    """Performs a circular right shift on the digits of x as per shift value and
     ↪ returns the result in string format.
    Returns reversed digits if shift exceeds number of digits.

    >>> circular_shift(12, 1)
    "21"
    >>> circular_shift(12, 2)
    "12"
    """
    x = str(x)
    # does not check shift exceeds string length and return reversed digits
    shift = shift  len(x)return x[-shift:] + x[:-shift]
```

Figure 25: GPT-4 failing solution on problem HUMANEVAL/65 in CONCISE

