# OpenReview forum: "Top Leaderboard Ranking = Top Coding Proficiency, Always? EvoEval: Evolving Coding Benchmarks via LLM"
_colmweb.org/COLM/2024/Conference — COLM_

### Official Review · Reviewer_brBL · 2024-04-24

**Rating:** 8
**Confidence:** 5
**Ethics Flag:** 1

**Summary:**

The paper introduces EvoEval, a novel benchmark suite for evaluating the coding abilities of large language models (LLMs) that addresses the limitations of existing benchmarks by evolving them into more varied and targeted domains. Through empirical analysis of 51 LLMs, the authors demonstrate that LLMs perform significantly worse on EvoEval compared to traditional benchmarks like HumanEval, with an average performance drop of 39.4%. This decrease, ranging from 19.6% to 47.7%, suggests potential overfitting to current benchmarks and results in significant changes in LLM rankings. Additionally, the paper highlights the brittleness of LLMs to changes in problem wording and underscores the importance of problem composition skills in programming tasks. EvoEval not only broadens the scope of evaluation but also provides a framework for continually adapting benchmarks in response to advancements in LLM capabilities.

**Questions To Authors:**

N/A

**Reasons To Accept:**

1. The paper addresses a critical need by highlighting that existing code generation benchmarks, such as HumanEval and MBPP, suffer from limited scale and diversity and are susceptible to data contamination.

2. The methodology is innovative; it creatively uses existing benchmarks and large language models (LLMs) to generate new, more diverse problems from those already in the benchmarks.

3. Figure 2 offers key insights by comparing LLM performance on HumanEval+ and EvoEval, revealing that LLMs fine-tuned for instruction-following may perform well on HumanEval+ due to data leakage, yet underperform on EvoEval.

4. The evaluation is thorough, assessing 51 LLMs across the new benchmark, providing a robust experimental section. The analysis of the results is detailed and insightful.

5. The appendix of the paper is detailed, providing comprehensive supplementary material that supports the findings and methodology.

**Reasons To Reject:**

1. The EvoEval benchmark leverages GPT-4 for problem creation, which is also used in instruct-finetuning datasets for some evaluated LLMs. This could lead to biased performance estimates for these models, and may also skew the results that some key findings in this paper is based on. It would be beneficial for the paper to clearly indicate which models have even been trained on GPT-4-generated data, perhaps using specific symbols or annotations.

2. Typo: "DeepSeeker" -> "DeepSeek-Coder"

---

> ### Author Rebuttal · Authors · 2024-05-29
>
> We thank the reviewer for the very detailed comments and suggestions on our work!
>
> > beneficial for the paper to clearly indicate which models have even been trained on GPT-4-generated data, perhaps using specific symbols or annotations.
>
> Thanks for this great suggestion to further improve the paper! We will definitely add this to the final version along with the additional discussion of potential biases.
>
> > Typo: "DeepSeeker" -> "DeepSeek-Coder"
>
> Thanks for catching this and we will fix it in the next revision.

---

> > ### Comment · Reviewer_brBL · 2024-05-31
> > **Thank you for your response**
> >
> > Thank you for your response and clarifications. I decide to keep my original review score.

---

### Official Review · Reviewer_AGkd · 2024-05-10

**Rating:** 7
**Confidence:** 4
**Ethics Flag:** 1

**Summary:**

This paper introduces a program synthesis benchmark called EVOEVAL for a comprehensive evaluation of LLM coding abilities. It aims to address the reliability limitation of previous benchmarks due to their potential data leakage issues. The proposed benchmark is created by evolving existing benchmarks into different targeted domains. In evaluation, it comprehensively considers 51 different LLMs across all benchmarks in EVOEVAL and the experiment results provide insightful and interesting findings.

**Questions To Authors:**

N/a

**Reasons To Accept:**

* This paper effectively tackles the shortcomings of prior code generation benchmarks by introducing a more comprehensive and robust benchmark to mitigate the potential data leakage problem. By doing so, it significantly improves the evaluation of LLMs' code generation capabilities across diverse aspects.  I would anticipate this benchmark will be widely adopted by the community.
* The experiments conducted are thorough and offer insightful findings.
* This paper is very well-presented.

**Reasons To Reject:**

N/a

---

> ### Author Rebuttal · Authors · 2024-05-29
>
> We truly appreciate the reviewer’s great comments! Please kindly let us know in case you have any questions later in the discussion period.

---

### Official Review · Reviewer_ihPF · 2024-05-10

**Rating:** 8
**Confidence:** 4
**Ethics Flag:** 1

**Summary:**

The current research piece introduces EvoEval, a new benchmark constructed by “evolving” (i.e. modifying) existing problems from HumanEval on different aspects, making them more difficult, more creative in their description, changing the problems in a subtle way, combining different problems into one or make the models to have (or produce) a helper function to solve the problem. In addition, two extra evolutions were presented, rephrasing problem descriptions, while preserving their semantics, by evolving in a more concise or in a more verbose way.
Problems were created using HumanEval as a starting point and prompting GPT-4 for their modification.
The resulting benchmark contains 828 problems across 7 different benchmarks. The benchmark was evaluated on 51 open and proprietary LLMs with coding capabilities.
Finding results interesting and new. While all the LLMs drop in performance, some of the best performing models on HumanEval drop significantly on EvoEval. Interestingly, some models have a significant drop in performance on subtle changes on their description.
The overall assessment of this work is good. This new benchmark provides new insights to the scientific community in regards to code synthesis while introducing a new way to assess the quality of such models.

**Questions To Authors:**

Icons from Figure 5 are pixelated

**Reasons To Accept:**

Strong work.

Clear, and easy to read.

A new benchmark bringing new insights to the scientific community.

Proper and extensive evaluation, always being compared with previous benchmarks, HumanEvan in this case.

**Reasons To Reject:**

No reasons to reject

---

> ### Author Rebuttal · Authors · 2024-05-29
>
> We thank the reviewer for their thoughtful comments! Please kindly let us know in case you have any questions later in the discussion period.
>
> > Icons from Figure 5 are pixelated.
>
> Thanks for catching this! We will ensure it is fixed for the final version of the paper.

---

### Official Review · Reviewer_Y48c · 2024-05-15

**Rating:** 8
**Confidence:** 4
**Ethics Flag:** 1

**Summary:**

This paper presents EvoEval, an evaluation benchmark for code generation tasks. The method converts existing problems into different tasks, with 7 different types of modifications.
The EvoEval benchmarks are challenge and the performance of many LLMs has dropped using the benchmarks. The paper further points out some interesting trends, such as the instruction-following LLMs are very sensitive even to subtle changes.
Overall, this paper identifies and attempts to solve a valuable problem: the contamination and potential overfit of LLMs on existing benchmarks. Theoretically, the method can be further used to create new benchmarks in the future.

*Writing*

Due to the relative complex setup and multiple types of “evol” methods, some explanations of the methods are a bit hard to follow. For example, the paper explains “COMBINE-NAÏVE” clearly, yet how “COMBINE”is done is a bit unclear.

**Questions To Authors:**

As mentioned in the text, models such as WizardCoder performs more robustly since it is trained on Evolved data. Would you consider models that are trained on similar evolution methods as a different form of overfitting?

**Reasons To Accept:**

- The benchmark presented by the paper is valuable to the community. Evaluation is one of the key components for advancement in AI.
- The paper presents interesting finding of the current landscape of the models, by performing experiments 51 different variations of LLMs. These trends, such as the ranking change, variations related to subtle changes, point out some potential weakness of the language models.
- The experiment is comprehensive and provide valuable insights

**Reasons To Reject:**

No particular reasons to reject.

---

> ### Author Rebuttal · Authors · 2024-05-29
>
> We appreciate the reviewer’s insightful question and recognition that our benchmark suite can be valuable to the community!
>
> > Unclear “COMBINE” dataset
>
> For our combined dataset, we first randomly select two HumanEval problems to combine together. In order to select problems that make sense to combine, we apply a simple heuristic
> to combine only problems of the same type together categorized based on the type of
> input arguments in the original problem. Our prompt used for the combined dataset is in Figure 12 (in Appendix D). We will work towards clarifying the benchmark creation process in more detail in the updated version of the paper.
>
> >models that are trained on similar evolution methods as a different form of overfitting?
>
> Thanks for this great question! We believe that models which are trained on evolving into more difficult problems like WizardCoder indeed contain a different form of overfitting. That is to say they overfit to the type of evolved problems they are trained on. For example, in our analysis, we found that while WizardCoder is better on more DIFFICULT problems, they perform worse on the COMBINE dataset since they are not trained to solve problems that combine multiple programming concepts.There actually motivated us to create a composite evolving dataset with different strategies for more comprehensive evaluation of code LLMs. We thank the reviewer again for this question and will add more discussion in the paper.

---

> > ### Comment · Reviewer_Y48c · 2024-06-05
> >
> > Thanks for answering the question. I missed the part that most evolution methods rely on prompting a language model, since the combine-naive method uses mainly concatenation.
> >
> > I guess a follow up question should've been whether or do we need to get around the need to have a model that can do the evolution.
> >
> > In terms of my rating, I would keep it the same regardless.

---

> > > ### Author Response · Authors · 2024-06-06
> > >
> > > Thanks for your reply!
> > >
> > > >I guess a follow up question should've been whether or do we need to get around the need to have a model that can do the evolution.
> > >
> > > Currently, we definitely need to use an LLM to perform the evolution. The reason is that many of the evolved problems require changing the semantic meaning of the original problem (e.g., making it more creative, or difficult). Doing this without an LLM would be challenging as we would need to design set of rules to perform these semantic transformations which cannot generalize to all sorts of input problems.The combine-naive is a special case where the simple rules can be applied without LLMs but in general we don't think its possible to go without using LLMs for the evolution.

---

### Decision · Program_Chairs · 2024-07-10

**Decision:**

Accept

**Comment:**

This paper introduces EvoEval, a novel benchmark designed to evaluate the coding abilities of large language models (LLMs) by addressing the limitations of existing benchmarks. EvoEval modifies existing problems from HumanEval across seven different dimensions to create more challenging and diverse tasks. The benchmark consists of 828 problems evaluated across 51 open and proprietary LLMs. The study reveals significant performance drops for LLMs on EvoEval compared to traditional benchmarks, highlighting issues of potential overfitting and sensitivity to subtle changes in problem descriptions. The results provide valuable insights into the robustness and reliability of current LLMs.

This paper addresses a critical need in the field of code generation by introducing EvoEval, a benchmark that mitigates the limitations of existing benchmarks such as HumanEval. EvoEval enhances evaluation by evolving existing problems into diverse and targeted domains, revealing significant performance drops and new insights into the brittleness of LLMs. The paper is well-written and presents a thorough experimental evaluation of 51 LLMs, providing a robust analysis of the results. While the methodology is innovative and the findings are valuable, the paper could benefit from addressing potential biases introduced by using GPT-4 for problem creation, especially given its role in training some of the evaluated LLMs. Despite this minor concern, the paper's contributions are substantial, and it is recommended for acceptance.